# CD26/DPP-4: Type 2 Diabetes Drug Target with Potential Influence on Cancer Biology

**DOI:** 10.3390/cancers13092191

**Published:** 2021-05-02

**Authors:** Emi Kawakita, Daisuke Koya, Keizo Kanasaki

**Affiliations:** 1Internal Medicine 1, Shimane University Faculty of Medicine, 89-1 Enya-cho, Izumo 693-8501, Japan; kawakita@med.shimane-u.ac.jp; 2Department of Diabetology & Endocrinology, Kanazawa Medical University, Uchinada 920-0293, Japan; koya0516@kanazawa-med.ac.jp; 3Division of Anticipatory Molecular Food Science and Technology, Medical Research Institute, Kanazawa Medical University, Uchinada 920-0293, Japan

**Keywords:** DPP-4, epithelial-mesenchymal transition, CXCL12, type 2 diabetes, metformin

## Abstract

**Simple Summary:**

Dipeptidyl peptidase (DPP)-4 inhibitor is widely used for type 2 diabetes. Although DPP-4/CD26 has been recognized as both a suppressor and inducer in tumor biology due to its various functions, how DPP-4 inhibitor affects cancer progression in diabetic patients is still unknown. The aim of this review is to summarize one unfavorable aspect of DPP-4 inhibitor in cancer-bearing diabetic patients.

**Abstract:**

DPP-4/CD26, a membrane-bound glycoprotein, is ubiquitously expressed and has diverse biological functions. Because of its enzymatic action, such as the degradation of incretin hormones, DPP-4/CD26 is recognized as the significant therapeutic target for type 2 diabetes (T2DM); DPP-4 inhibitors have been used as an anti-diabetic agent for a decade. The safety profile of DPP-4 inhibitors for a cardiovascular event in T2DM patients has been widely analyzed; however, a clear association between DPP-4 inhibitors and tumor biology is not yet established. Previous preclinical studies reported that DPP-4 suppression would impact tumor progression processes. With regard to this finding, we have shown that the DPP-4 inhibitor induces breast cancer metastasis and chemoresistance via an increase in its substrate C-X-C motif chemokine 12, and the consequent induction of epithelial-mesenchymal transition in the tumor. DPP-4/CD26 plays diverse pivotal roles beyond blood glucose control; thus, DPP-4 inhibitors can potentially impact cancer-bearing T2DM patients either favorably or unfavorably. In this review, we primarily focus on the possible undesirable effect of DPP-4 inhibition on tumor biology. Clinicians should note that the safety of DPP-4 inhibitors for diabetic patients with an existing cancer is an unresolved issue, and further mechanistic analysis is essential in this field.

## 1. Introduction

Type 2 diabetes mellitus (T2DM) is associated with an increased risk of certain cancers and poor prognosis [1,2]. Mechanistically, hyperinsulinemia, insulin resistance, hyperglycemia, dyslipidemia, and chronic inflammation may promote cancer progression in T2DM patients. Moreover, the long-term use of anti-diabetic agents also potentially influences tumor behavior in diabetic patients with cancer [3,4]. Although large clinical trials to determine the safety of anti-diabetic drugs for cardiovascular outcome in diabetic patients have been performed, such a clinical study usually did not include those patients already known to have cancer, and observational intervals of trials to verify the impact of anti-diabetic drugs on cancer progression are rare. Therefore, the safety profile of anti-diabetic agents on cancer-bearing diabetic patients has never been thoroughly evaluated.

Dipeptidyl peptidase (DPP)-4 inhibitors are widely used for treatment in T2DM patients. DPP-4 is characterized as a T-cell differentiation antigen (CD26) and plays a multifunctional role through its enzymatic and nonenzymatic action. DPP-4 cleaves many substances, not only incretin hormones; DPP-4 inhibitors potentially increase many growth factors and chemokines that may induce cancer progression. Besides this, DPP-4 has many pivotal roles in immune function, inflammation, and antioxidative response. Therefore, the pleiotropic effects of DPP-4 inhibitors would not always be favorable, especially in cancer-bearing T2DM patients.

Since other authors have mentioned the favorable effects of DPP-4 suppression for cancer biology elsewhere [5,6,7,8,9,10], this review mainly focuses on the undesirable effects of DPP-4 inhibition in T2DM patients with cancer, including our preclinical data.

## 2. The DPP-4 Gene Family

This review fundamentally focused on DPP-4 for the Special Issue “CD26 and cancer” in this journal. However, introducing the general aspect of the biology of the DPP-4 gene family on cancer could help to understand the whole picture of this interesting peptidase family regarding cancer biology. Details are fully described elsewhere [11,12,13,14].

The DPP-4 gene family, a subgroup of the prolyl oligopeptidase family of enzymes, specializes in the cleavage of prolyl bonds. Among its family members, DPP-4, DPP-8, DPP-9, and fibroblast activation protein (FAP) display enzymatic action. The specific substrate or pathophysiological significance on cancer of DPP-8 and DPP-9 are not elucidated yet, but some tumor-promoting effects of either DPP-8 or DPP-9 have been reported [15,16]. At the least. the inhibition of DPP-8 or DPP-9 could potentially influence the immune system and differentiation of certain cells. FAP is a very closely related enzyme to DPP-4 and exhibits higher levels in some cancers, especially in cancer-associated fibroblast. Indeed, the DPP-4 inhibitor linagliptin inhibited FAP as well [17]. 

### DPP-4

DPP-4/CD26 is a cell surface glycoprotein expressed in various types of cells and tissues. The expression level of DPP-4 is incredibly high in the kidneys and small intestine. There are two forms of DPP-4: membrane-bound and soluble. The transmembrane protein domain of DPP-4 is anchored to the cell membrane, and its cytoplasmic domain is located at the N-terminus. The extracellular lesion of DPP-4 contains highly glycosylated, cysteine-rich, and catalytic regions. The C-terminal loop of DPP-4 is vital for catalytic efficacy and dimerization [18]. A soluble form of DPP-4 (sDPP-4) is cleaved off the membrane by matrix metalloproteases [19] and exists in serum and other bodily fluids.

DPP-4 plays diverse biological functions through enzymatic-dependent and independent actions. For the enzymatic action, DPP-4 digests many polypeptides, including chemokines, neuropeptides, and incretin hormones such as glucagon-like peptide (GLP)-1 and glucose-dependent insulinotropic peptide [20]. Increased incretin hormones by DPP-4 inhibition leads to the glucose-dependent secretion of insulin; thus, the DPP-4 inhibitor is a widely prescribed drug for T2DM in clinics. DPP-4 also has molecular functions via interactions with adenosine deaminase, extracellular matrix (ECM) proteins, and caveolin-1 for non-enzymatic action. The roles on co-receptor activity for viral entry, cell proliferation and migration of DPP-4 are also reported [21,22,23,24,25]. The membrane-bound form of DPP-4 is also known to have a pivotal role in the interactions between integrin β1 and ECM [26], by which integrins regulate the cytoskeletal organization and intracellular-signaling pathways [27,28]. We have reported that DPP-4 plays a profibrotic role through interaction with integrin β1, and subsequently induces endothelial-mesenchymal transition [29].

In human serum, sDPP-4 plays a central role in enzymatic activity [30,31]. sDPP-4 induces cell signaling and human lymphocyte proliferation [32], independent of its catalytic action. In human adipocytes, skeletal muscle cells, and smooth muscle cells, sDPP-4 inhibits Akt activation [33]. The process of sDPP-4 synthesis is still not completely understood. Lamers et al. indicated that sDPP-4 is released from differentiated adipocytes more than preadipocytes, and that human adipocytes-secreted tumor necrosis factor-α and insulin increase both the levels of sDPP-4 and DPP-4 mRNA [33]. Others reported that the major sources of sDPP-4 are adipocytes and the liver. Additionally, DPP-4 inhibition increases the levels of sDPP-4 in plasma from endothelial or hematopoietic cells [34].

Since the diverse biological functions of DPP-4, cancer-promoting cytokine/ chemokine modulation and ECM interaction would be relevant to cancer cell behavior, DPP-4 inhibitors also potentially influence cancer cell biology beyond blood glucose control. Its biological effects are complex and different, depending on tumor types and the microenvironment.

## 3. DPP-4 Expression in Primary Tumors

DPP-4 is expressed in many solid tumors or hematologic malignancies (Table 1). The overexpression of DPP-4 in several cancers induces an anti-tumor effect [35,36]. DPP-4 in the primary tumor may have an essential role in regulating cancer biology, and DPP-4/CD26 has also been noted as a potential biomarker of tumor progression. Mezawa et al. demonstrated that the decreased expression of CD26 is associated with poor prognosis in breast cancer patients. Furthermore, this decreased CD26 expression is attenuated by the suppression of transforming growth factor (TGF)-β and C-X-C motif chemokine 12 (CXCL12) [37]. We also showed that DPP-4 ablation in a primary mammary tumor induced tumor growth and metastasis via the CXCL12-mediated pathway (Figure 1) [38]. In small cell lung cancer, DPP-4 levels were significantly diminished; however, restoration of DPP-4 induced apoptosis and cell cycle arrest via the p21 accumulation, and suppressed tumor growth and metastasis [39].

On the other hand, an increased CD26 expression level has been associated with poor prognosis in pancreatic cancer patients [46] and other cancers [40,50]. DPP-4 expression in the primary tumor may regulate cancer behavior; however, the expression of DPP-4 in the primary tumor appears to be the heterogeneity patterns, depending on tumor type, stage, microenvironment, and host condition. It was also reported that DPP-4 expression and its secreted activity are uncoupled under hypoxia in ovarian cancer cells via the shedding of inactive DPP-4 from ovarian cancer cells [53]. Although several reports have suggested that the overexpression of DPP-4 would be a potential tumor prognosis marker in certain cancers, it is unresolved (1) how diverse functions of DPP-4 influence the cancer progression, and (2) whether DPP-4 inhibition is relevant as a therapy for patients with such DPP-4 overexpressed cancers. In addition, most reports about the correlation between DPP-4 expression in tumor and cancer prognosis did not mention whether patients had diabetic conditions or not. How the use of DPP-4 inhibitors impacts the DPP-4 expression of cancer in diabetic patients is not yet elucidated. It is essential to note that the clinical biomarkers are not always attributable to their biological importance in the disease activity; careful interpretation would be required.

## 4. The Impact of DPP-4 Inhibition on Existing Tumors

Several preclinical studies have addressed the potential link between DPP-4/CD26 and tumor progression. However, the mechanistic consequences of DPP-4 inhibition for a cancer-bearing individual are still not completely understood. Some studies described that the DPP-4 inhibitor is correlated with cancer cell invasiveness, metastasis and chemotherapy resistance in certain cancers. An early study by Beckenkamp et al. showed that the DPP-4 inhibitor sitagliptin increases cell migration and adhesion in cervical cancer cells [54]. Wang et al. also reported that the DPP-4 inhibitor saxagliptin and sitagliptin activate nuclear factor E2-related factor 2 (NRF2)-mediated antioxidant response, resulting in promoting metastasis of multiple tumors [55]. In papillary thyroid carcinoma cells, saxagliptin also promotes cancer cell migration and invasion through upregulation of NRF2 [56]. However, oxidative stress is known to suppress [57] or to induce [58] cancer invasion and metastasis. Redox balance plays an intricate role in cancer biology; DPP-4 inhibitors could also influence the tumor progression both favorably and unfavorably according to conditions.

For an alternative mechanism, DPP-4 inhibitors increase their substrate levels: neuropeptides, growth factors, cytokines, and chemokines. An elevated level of these substrates of DPP-4 can potentially influence cancer progression (Table 2).

CXCL12, and its receptor C-X-C receptor 4 (CXCR4), are related to hematopoiesis, angiogenesis, stem cell homing, and tumor progression and behavior [59,60]. Sun et al. reported that DPP-4 regulates prostate cancer metastasis in vitro and in vivo by the degradation of CXCL12 [61]. We also found that the DPP-4 inhibitor induces breast cancer metastasis via the inhibition of CXCL12 degradation. Accumulated CXCL12 interacted CXCR4 and contributed to CXCL12/CXCR4-mediated epithelial-mesenchymal transition (EMT) induction (Figure 1) [62]. Interestingly, EMT induction by DPP-4 inhibitor was independent of TGF-β signaling. DPP-4 inhibitor-induced breast cancer metastasis was attenuated by the CXCR4 inhibitor AMD3100, with the suppression of genes associated with EMT and a mammalian target of rapamycin (mTOR) signaling pathway in the primary tumor [38,62]. The mTOR pathway is important for driving EMT processes in a mammary tumor [63,64], and mTOR inhibition suppresses breast cancer proliferation, migration and metastasis [65,66,67]. In addition, some reports described the induction of the mTOR signaling pathway by DPP-4 inhibition [62,68,69]. In this regard, the activation of the mTOR pathway may be one of the critical factors for EMT induction by DPP-4 inhibition (Figure 1). However, in contrast, Varela-Calviño et al. recently reported that sitagliptin suppressed the metastatic potential of colorectal cancer cells in vitro [10]. Nevertheless, the effects of DPP-4 inhibitors on tumor metastasis remain unknown and further investigation is important.

EMT is also relevant for chemotherapy resistance, not only for invasion or metastasis. Cells undergoing EMT processes are known to induce ATP-binding cassette (ABC) transporter overexpression, which is related to cancer drug resistance in several cancer cells [94,95]. With regard to this, we have reported that the DPP-4 inhibitor increases ABC transporters via EMT induction, resulting in breast cancer chemoresistance (Figure 1) [76]. A xenograft model of DPP-4-overexpressed epithelial ovarian cancer cells exhibited as larger bodies, but at the same time, displayed more chemosensitivity to paclitaxel [43]. Nevertheless, due to such heterogeneities in these preclinical studies, we should be carefully monitoring some potential influences, especially the unfavorable ones, of DPP-4 inhibitors in diabetic patients on cancer chemotherapy. Further mechanistic validation would be required.

## 5. The Influence of DPP-4 Inhibitors on Cancer: Clinical Evidence

Based on the current evidence from large clinical trials, DPP-4 inhibitors are safe for human health without significant concern about new onsets of cancers. Four cardiovascular outcomes trials (CVOTs) of DPP-4 inhibitors in T2DM patients represented that DPP-4 inhibitors (saxagliptin, alogliptin, sitagliptin, and linagliptin) had no statistically significant impact on cancer incidence, yet some trends of increased incidence of certain cancers are described (Table 3) [96,97,98,99,100]. The median follow-up periods were also short in order to evaluate the de novo carcinogenetic effect of DPP-4 inhibitors, and the number of cancer cases was limited in these trials (Table 3). Moreover, it is unknown whether DPP-4 inhibitors influence existing tumor growth, metastasis, and chemotherapy resistance because these trials excluded cancer-bearing diabetic patients.

Previous preclinical studies indicated that DPP-4 inhibitors could accelerate tumor growth and metastasis, not only the potential cancer-protective effects. Therefore, a prospective randomized trial to test the safety of DPP-4 inhibitors on cancer-bearing diabetic patients will never be performed. We cannot have a clear answer yet; however, clinicians should recognize the theoretical potential risks when prescribing DPP-4 inhibitors in cancer-bearing T2DM patients.

### 5.1. The Impact of DPP-4 Inhibitors on Cancer Incidence in Diabetic Patients

Observational studies and retrospective studies have discussed the link between DPP-4 inhibitor use and cancer incidence in diabetic patients. DPP-4 inhibitor is one of the incretin-based drugs, by which the blood glucose-dependent secretion of insulin from pancreatic β cells regulates blood glucose levels, and therefore is associated with a low risk of hypoglycemia with monotherapy. There has been some concern about an increased risk of certain cancers by incretin drugs, especially pancreatic cancer and thyroid cancer [102,103,104,105]; in contrast, later publications denied such a trend [106,107,108,109,110,111,112,113,114]. However, another retrospective study indicated incretin-based drugs are associated with an increased risk of pancreatic cancer with the adjusted hazard ratio (aHR) 2.14 (95%CI: 1.71–2.67), compared with other oral antidiabetic drugs [115]. More recently, Lee et al. also reported that DPP-4 inhibitor users have a 50% increased risk of pancreatic cancer incidence (aHR 1.50, 95%CI: 1.02–2.20) among newly diagnosed T2DM patients. It is likely that such a risk was increased during the early exposure period of DPP-4 inhibitor prescription [116]. Sitagliptin also increased the risk of thyroid cancer only in the first year of its use [105]. These data suggested that DPP-4 inhibitors induce existing microtumor/precancerous cell visibility rather than inducing de novo carcinogenesis. On the other hand, a propensity-matched cohort study of T2DM patients in Korea represented that DPP-4 inhibitor treatment did not increase the risk of pancreatic and thyroid cancer compared to metformin treatment [117].

Abrahami et al. [118] showed the association between incretin-based drugs and an increased risk of cholangiocarcinoma in T2DM patients. In this study, the incidence of cholangiocarcinoma is significantly increased by 77% in DPP-4 inhibitor users (HR 1.77, 95%CI: 1.04–3.01), and an increasing trend in GLP-1 receptor agonist (GLP-1RA) users as well (HR 1.97, 95%CI: 0.83–4.66). The undesirable effect of DPP-4 inhibitors remained significant after adjusting with sulfonylurea users [118]. These drugs increase the plasma levels of GLP-1 or GLP-1RA, which may stimulate cholangiocarcinoma proliferation and anti-apoptotic response [119,120,121]. In addition to the elevated level of endogenous GLP-1, DPP-4 inhibitor administration increases numerous substrates that also influence the tumor microenvironment and the immune system, which are essential for tumor onset. Cholangiocarcinoma is a rare cancer type; therefore, this could be the result of prescription biases. However, also such a finding could be a sign of the risk of DPP-4 inhibitors in certain cancers, and it should not be overlooked. In contrast, DPP-4 inhibitor users had no elevated risk of colorectal cancer incidence in diabetic patients [122], and that the DPP-4 inhibitor was also associated with a significantly reduced risk of colorectal cancer among T2DM patients [123].

As mentioned above, there are several studies about the relationship between DPP-4 inhibitors and site-specific cancer incidence. Based on the available clinical reports, however, the incidence of cancer is not significantly related to the DPP-4 inhibitor prescription when adjusted by cofounders [124]. A meta-analysis of randomized clinical trials showed that DPP-4 inhibitors were not related to an increased risk of developing cancers compared to a placebo or other anti-diabetic drugs in T2DM patients [125]. Other meta-analyses also reported that DPP-4 inhibitors did not increase the risk of cancers of the digestive system [126] or overall cancers [123]. The conclusion by clinical analysis when examining the association between DPP-4 inhibitor and cancer incidence is influenced by diverse factors. Firstly, the follow-up period was short for validating the impact of anti-diabetic drugs on cancer incidence. Secondly, the number of patients using DPP-4 inhibitors alone was limited. Patients who are prescribed DPP-4 inhibitors as a second- or third-line anti-diabetic drug often display poor blood glucose control, which potentially promotes cancer initiation [127]. Nevertheless, in the information obtained from these observational, retrospective, or cohort studies, complete elimination of significant biases would never be possible; therefore, careful consideration is required to interpret these data. Both a preclinical study and larger clinical studies with a longer term are needed to confirm the safety of DPP-4 inhibitors for cancer incidence.

### 5.2. The Influence of DPP-4 Inhibitors on Cancer Prognosis and Metastasis

SEER- and Medicare-linked database analysis indicated that T2DM patients taking a DPP-4 inhibitor had a worse trend of overall survival (OS) with HR 1.07 (95%CI: 0.93–1.25, *p* = 0.33) in the breast cancer cohort and HR 1.07 (95%CI: 0.93–1.24, *p* = 0.68) in the pancreatic cancer cohort, whereas DPP-4 inhibitor use had significant OS benefit with HR 0.77 (95%CI: 0.64–0.93, *p* = 0.005) in patients with prostate cancer [128]. Ali et al. reported DPP-4 inhibitor had a positive effect on progression-free survival in patients with advanced airway and colorectal cancers and diabetes by a multicenter retrospective analysis [129].

Cancer metastasis and chemotherapy resistance are important factors for the prognosis of cancer patients. Preclinical studies have reported that DPP-4 inhibitors potentially increase the risk of tumor metastasis [55]. A cohort study in Korea represented that the DPP-4 inhibitor increases newly onset metastasis of primary thyroid cancer in diabetic patients [130]. Rathmann et al. performed the observational study, utilizing a database of ICD code filled in by primary care physicians, to analyze the association between DPP-4 inhibitor therapy and the risk of metastasis in T2DM patients with newly diagnosed breast, prostate and digestive organ cancers [131]. In this observational study, DPP-4 inhibitors did not increase a risk of metastasis in T2DM patients with breast (aHR 1.00, 95%CI: 0.49–2.02), prostate (0.98, 0.54–1.77) or digestive organ cancers (0.97, 0.57–1.66) [131].

As with the association between DPP-4 inhibitor use and newly diagnosed cancer, the conclusions on the effect of DPP-4 inhibitors on cancer prognosis and metastasis differed with factors such as cancer type or patient profile. Nevertheless, the currently available clinical study indicated that there is no significant risk of a cancer prognosis and metastasis among cancer-bearing diabetic patients on DPP-4 inhibitor therapy. Importantly, the retrospective analysis gives us some hypotheses, but the absence of information about cancer characteristics, such as its stage or receptor expression, makes it difficult to interpret these data. On top of that, diabetes is an independent risk factor for cancer progression, thus epidemiological studies investigating the association between anti-diabetic drugs and cancer outcomes come with diverse, unavoidable limitations. Recently, Vihervuori et al. reported that the use of an anti-diabetic agent, especially metformin and insulin, was associated with the worse prognoses in patients with prostate cancer [132]. This result may suggest that the presence of diabetes with metabolic abnormalities such as hyperglycemia or insulin resistance displays stronger influences on prostate cancer progression, rather than specific drug use. Indeed, hyperglycemia and/or insulin resistance are linked to a worse prognosis of cancer patients [133,134,135]. Therefore, it is challenging to evaluate the effect of specific anti-diabetic drugs on cancer progression in diabetic patients with retrospective analysis.

Furthermore, a clinical report exploring the impact of DPP-4 inhibitors on chemotherapy outcomes in diabetic patients with cancer is lacking. In previous preclinical studies, either EMT or increased CXCL12 level, the condition associated with DPP-4 inhibition, is relevant for chemoresistance [76,136,137]. DPP-4 also plays an important role in the immune system, especially T-cell functions. A clinical trial investigating the outcome of chemotherapy, including immune-checkpoint inhibitors, for cancer-bearing diabetic patients with DPP-4 inhibitors is also required.

## 6. Perspective: Co-Prescription with Metformin

We have recently reported that metformin mitigates DPP-4 inhibitor-induced breast cancer EMT and metastasis via the suppression of mTOR activation [38]. Metformin is the first-line drug for T2DM, and thus, metformin is often prescribed with other anti-diabetic drugs. Metformin has been also recognized as an anti-cancer agent for some time; metformin users among diabetic patients displayed a decreased risk of colorectal, pancreatic and breast cancer compared to metformin non-users [138,139]. Metformin also exhibits a protective effect on overall cancer morbidity and mortality [140]. Additionally, some reports indicated the possibility that metformin impacts the cancer-related outcome on DPP-4 inhibitor use in diabetic patients. Retrospective analysis data represents that breast cancer-bearing diabetic patients taking metformin had significant OS benefit either with or without DPP-4 inhibitors, whereas DPP-4 inhibitors alone displayed a worse trend [128]. In this regard, DPP-4 inhibitor treatment among T2DM patients with colorectal and lung cancers represents a significant survival advantage alongside metformin, whereas DPP-4 inhibitor treatment alone did not reach a significant survival benefit [7].

Regarding the effect on cancer metastasis, the co-prescription of metformin with DPP-4 inhibitors could prevent an elevated risk of thyroid cancer metastasis with DPP-4 inhibitor alone in T2DM patients [130]. Noh et al. reported that using a DPP-4 inhibitor alone increases the metastasis of primary thyroid cancer with HR 3.89 (95%CI: 1.04–9.64) in T2DM patients, whereas a combination with metformin attenuated such an undesirable effect of DPP-4 inhibitor (HR 0.75, 95%CI: 0.57–1.01) [130]. This clinical evidence suggests the effect of a DPP-4 inhibitor on cancer progression in T2DM patients with certain cancers differs depending on whether it is co-prescribed with metformin. However, Matthews et al. reported that in their analysis, even under a combination therapy with metformin, DPP-4 inhibitor vildagliptin prescription increased the trend of prostate and breast cancer incidence when compared to metformin alone (Table 3) [101].

Interestingly, our preclinical data demonstrated that metformin represents anti-tumor/anti-metastatic effect only when prescribed with a DPP-4 inhibitor; nevertheless, metformin alone had no impact on basal breast cancer proliferation and metastasis in vivo [38]. Indeed, some randomized trials showed no impact of metformin on breast cancer progression and prognosis [141,142,143], even though several preclinical data have reported the anti-tumor effect of metformin in breast cancer. Further mechanistic investigation is definitely needed, using metformin as an anti-tumor agent in a clinical setting. Metformin may preferentially target mesenchymal cells or stem cells [144,145,146], indicating some clue as to the beneficial effects of metformin on human health. DPP-4 inhibition advances the EMT process of cancer cells to a more mesenchymal phenotype, relating to cancer migration, invasiveness, metastasis, chemoresistance, or stemness. Therefore, metformin may have more anti-tumor influence on a mammary tumor when co-prescribed with DPP-4 inhibitors, compared with metformin alone (Figure 2). Because DPP-4 inhibitors are often prescribed with other anti-diabetic drugs, the differences induced by the interaction of drugs are also of interest.

## 7. Conclusions

As DPP-4/CD26 has various vital functions in cancer biology, DPP-4 inhibitors potentially act as either suppressors or inducers in cancer development. Available clinical evidence has indicated no clear association as yet between DPP-4 inhibitors and cancer incidence or prognosis in diabetic patients. However, importantly, the safety profile of a DPP-4 inhibitor (the same as other anti-diabetic drugs) on cancer progression of existing cancer or recurrence has not yet been established. Thus, further mechanistic investigations about the link between DPP-4 inhibitors and cancer biology, especially in diabetic conditions, are an essential research topic in both diabetology and oncology.

## Figures and Tables

**Figure 1 cancers-13-02191-f001:**
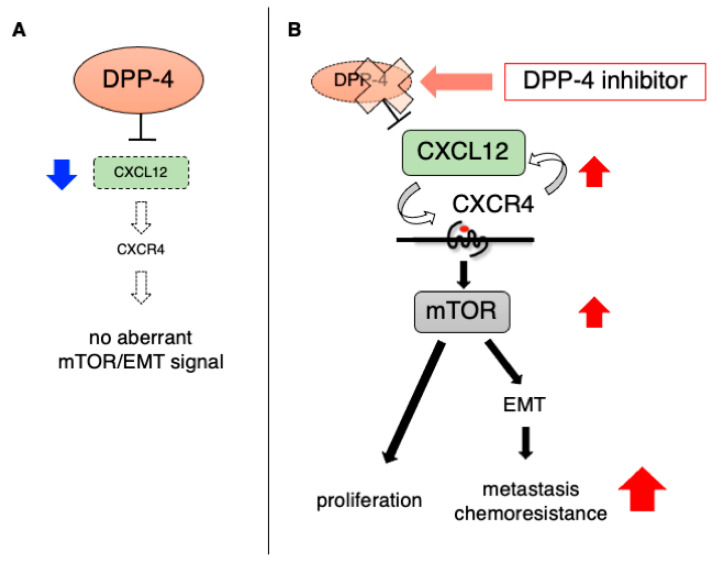
The impact of DPP-4 inhibitor on mammary tumor via CXCL12/CXCR4 downstream pathway. (**A**) DPP-4 digests CXCL12 for enzymatic action, thus the CXCL12-mediated CXCR4 downstream pathway in cancer is not strongly activated in the presence of DPP-4. (**B**) The DPP-4 inhibitor suppresses the degradation of CXCL12 and increases the level of CXCL12. Elevated CXCL12 interacts with its receptor CXCR4 and induces mTOR activation. The DPP-4 inhibitor-induced CXCL12/CXCR4/mTOR pathway causes mammary tumor proliferation. The activation of mTOR also causes EMT, which induces metastasis and chemoresistance in the mammary tumor. CXCL12: C-X-C motif chemokine 12; CXCR4: C-X-C receptor 4; DPP-4: dipeptidyl peptidase-4; EMT: epithelial-mesenchymal transition; mTOR: mammalian target of rapamycin.

**Figure 2 cancers-13-02191-f002:**
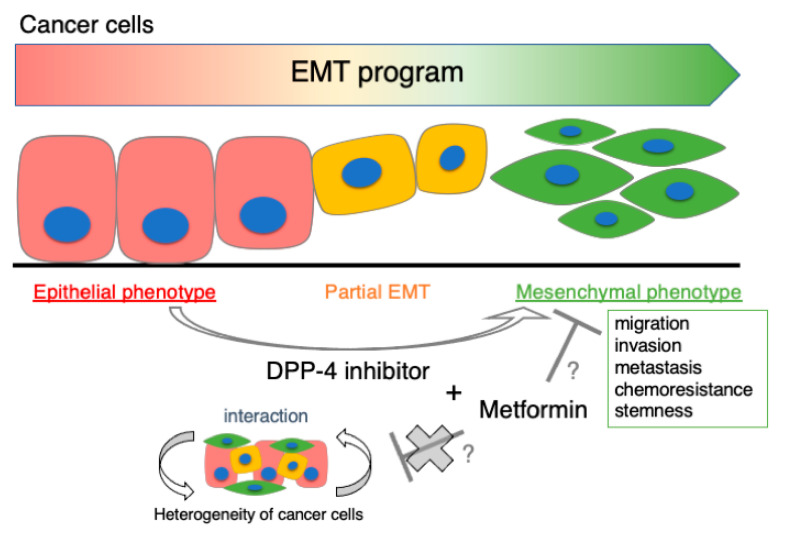
DPP-4 inhibitor-induced cancer progression is attenuated by metformin. Cancer cells exist in various phases of the EMT process; epithelial-like, mesenchymal-like, or partial phenotype of cancer cells. These heterogeneous cancer cell populations interact with each other and effectively develop in cancer biology. The DPP-4 inhibitor progresses cancer cells to a more mesenchymal phenotype, which is associated with tumor progressions such as migration, invasion, metastasis, chemoresistance and stemness. Metformin is likely to have an anti-tumor effect on homogenous mesenchymal cancer cell population induced by DPP-4 inhibitor. DPP-4: dipeptidyl peptidase-4; EMT: endothelial-mesenchymal transition.

**Table 1 cancers-13-02191-t001:** DPP-4/CD26 expression in primary tumors and its role in cancer progression processes.

Tumor Type	Possible Association Between DPP-4/CD26 Expression in Tumors and Tumor Progression	Reference
Breast cancer	Decreased stromal CD26 expression in tumors is associated with poor outcomes for BC patients.	[37]
	DPP-4 knockdown in a primary mammary tumor induces tumor growth and metastasis in vivo.	[38]
Colorectal cancer	High expression levels of CD26 in tumors is associated with distant metastasis and worse overall survival in CRC patients.	[40]
	Expression of stromal CD26 after preoperative CRT is associated with tumor recurrence and prognosis in rectal cancer patients.	[41]
Hepatocellular carcinoma	Low levels of DPP-4 expression in tumors are linked to the aggressiveness of HCC and poor overall survival in HCC patients.	[42]
Lung cancer	Restoration of DPP-4 expression in NSCLC cells contributes to inhibiting cell progression in vitro and in vivo.	[39]
Ovarian cancer	DPP-4 overexpression contributes to prolonged survival, decreased invasive activity, and increased chemosensitivity in vitro and in vivo.	[43,44]
	DPP-4 expression is associated with lymph node metastasis and a worse stage in tumor samples of ovarian cancer patients.	[45]
Pancreatic tumor	CD26 expression is significantly increased in tumors and its level correlates with overall survival in PDAC patients.	[46]
Prostate cancer	CD26 expression in prostate cancer tissues is correlated with CXCR4, PSA level, tumor residue, cancer stage, and tumor size.	[47]
Thyroid cancer	CD26 expression is negatively correlated with GLP-1R expression and has no significant association with the survival of patients with medullary thyroid carcinoma.	[48]
Urothelial carcinoma	DPP-4 overexpression in tumors is associated with the clinical aggressiveness of UCs.	[49]
Hematological malignancies	CD26 expression correlates with a poor response to 2′-deoxycoformycin in T-cell leukemia/lymphomas.	[50]
	CD26 expression is associated with an unfavorable clinical outcome in B-CLL patients.	[51]
	CD26 expression on B-CLL cells is associated with the tumor mass and influences time to treatment.	[52]

BC, breast cancer; BCLL, B-cell chronic lymphocytic leukemia; CRC, colorectal cancer; CRT, chemoradiotherapy; HCC, hepatocellular carcinoma; NSCLC, non-small cell lung cancer; PDAC, pancreatic ductal adenocarcinoma; UC, Urothelial carcinoma.

**Table 2 cancers-13-02191-t002:** Substrates of DPP-4/CD26 and potential downstream effect on tumor progression.

Substrates	Consequences of Processing by DPP-4	Receptor Recognition	Possible Activity of Substrates in Tumor Progression	Reference
Substance P	inactivation	-	promotes pancreatic cancer cell proliferation and invasion	[70]
Neuropeptide Y	inactivation	-	promotes myeloid cell infiltration and increases IL-6 levels in prostate cancer cell	[71]
CXCL9/Mig	inactivation	CXCR3 decreased	enhances breast cancer cell invasiveness/migration and reduces immune cell infiltration	[72,73]
CXCL10/IP-10	inactivation	CXCR3 decreased	enhances breast cancer cell invasiveness/migration and reduces immune cell infiltration	[72,73]
			promotes colorectal carcinoma cell invasiveness	[74]
			promotes gastric cancer cell invasion	[75]
CXCL11/I-TAC	inactivation	CXCR3 decreased	enhances breast cancer cell migration	[73]
CXCL12/SDF-1	inactivation	CXCR4 decreased	induces breast/mammary cancer cell migration, metastasis and chemoresistance	[38,62,76]
			induces prostate cancer cell invasion and metastasis	[61]
			effects on endometrial adenocarcinoma cell proliferation	[77]
CCL3/LD78β	inactivation	CCR3 decreased	promotes migration and invasion of esophageal squamous cell carcinoma	[78]
	activation	CCR1/CCR5 increased	induces proliferation and invasion of oral squamous cell carcinoma	[79]
CCL5/RANTES	inactivation	CCR1/CCR3 decreased	promotes breast cancer migration, invasion, metastasis and recurrence	[80,81,82]
	activation	CCR5 increased	promotes prostate cancer cell migration and metastasis	[83]
			promotes pancreatic cancer cell invasion and migration	[84]
CCL11/eotaxin	inactivation	CCR3 decreased	promotes the proliferation, migration and invasion of glioblastoma cells	[85]
CCL14/HCC-1	inactivation	CCR1/CCR3/CCR5 decreased	potential prognostic biomarker of hepatocellular and ovarian cancer	[86,87]
			promotes angiogenesis and metastasis of breast cancer	[88]
			suppress the tumor progression of colorectal and hepatocellular carcinoma	[89,90]
CCL22/MDC	inactivation	CCR4 decreased	increases oral cancer cell proliferation, invasion and migration	[91]
			stimulates the migration of Tregs and impairs antitumor immunity in ovarian cancer	[92]
			promotes lymph node metastasis of CCR4+ head and neck squamous cell carcinoma	[93]

Mig, monokine-induced interferon-γ; IP-10, interferon-γ-inducible protein 10; I-TAC, interferon-inducible T-cell chemo-attractant; SDF-1, stromal cell-derived factor 1; HCC-1, hemofiltrate CC chemokine-1; MDC macrophage-derived chemokine; IL-6, interleukin-6.

**Table 3 cancers-13-02191-t003:** Cancer incidence in large clinical trials with DPP-4 inhibitors.

	SAVOR-TIMI 53 [96,100]	EXAMINE [97]	TECOS [98]	CARMELINA [99]	VERIFY [101]
DPP-4 inhibitor	Saxagliptin	Alogliptin	Sitagliptin	Linagliptin	Vildagliptin + metformin
Follow-up (years)	2.1	1.5	3.0	2.2	up to 5
Patients	16,492	5380	14,671	6979	2001
Age (years)	65.1	61.0	65.5	66.1	54
Body-mass index ⁑—mean	31.1	28.7	30.2	31.4	31.2
Glycated hemoglobin—mean (%)	8.0 ± 1.4	8.0 ± 1.1	7.2 ± 0.5	7.9	6.7
Duration of diabetes (years)	10.3	7.1	11.6	15.0	3.3 (months)
Metformin use—no. (%)					
Placebo—no. (%)	5684 (69.2)	1805 (67.4)	6030 (82.2)	1927 (55.3)	all
DPP-4 inhibitors—no. (%)	5789 (69.9)	1757 (65.0)	5936 (81.0)	1881 (53.8)	all
Inclusion criteria	Established CVD, multiple risk factors for VD	An ACS with 15 to 90 days before randomization	Established CVD	High CV and renal risk	Diagnosed for type 2 diabetes within 2 years prior to enrolment
Primary outcome	3P-MACE	3P-MACE	4P-MACE	3P-MACE	The time from randomization to initial treatment failure (*)
Incidence of cancer			**		
Placebo—no. (%)	362 (4.4)	51 (1.9)	371 (5.1)	134 (3.8)	54 (5.4) ※
DPP-4 inhibitors—no. (%)	327 (3.9)	55 (2.0)	341 (4.7)	116 (3.3)	62 (6.2) ※※
P value	0.15	0.77	unknown	unknown	unknown
Pancreatic cancer		no reports			
Placebo—no. (%)	5	-	10 (0.1)	4 (0.1)	2 (0.3) ※
DPP-4 inhibitors—no. (%)	12	-	9 (0.1)	11 (0.3)	3 (0.3) ※※
Other cancers—no. (%)	unknown	unknown	unknown	Colon cancer	Prostate cancer
				Placebo: 8 (0.2)	Met: 0
				Linagliptin: 6 (0.2)	Vildagliptin + Met: 6 (0.6)
				Gastric cancer	Breast cancer
				Placebo: 3 (0.1)	Met: 1 (0.1)
				Linagliptin: 0	Vildagliptin + Met: 3 (0.3)

ACS: acute coronary syndrome; CV: cardiovascular; CVD: cardiovascular disease; VD: vascular disease; Met: metformin. 3P-MACE: a composite of cardiovascular death, nonfatal myocardial infarction or nonfatal stroke; 4P-MACE: a composite of 3P-MACE plus hospitalization for unstable angina. ⁑ Body-mass index is the weight in kilograms divided by the square of the highest in meters. * defined as HbA_1c_ measurement of at least 53 mmol/mol (7.0%) at two consecutive scheduled visits. ** including benign or unspecified neoplasm. ※ Monotherapy group (metformin); ※※ Combination therapy group (vildagliptin + metformin).

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
