# Peer review of "CD26/DPP-4: Type 2 Diabetes Drug Target with Potential Influence on Cancer Biology"

_cancers, 2021, doi:10.3390/cancers13092191_

Round 1

Reviewer 1 Report

This manuscript is a narrative review focused on the potential deleterious effects of DPP-4 inhibitors in type 2 diabetic patients with cancer. The authors describe the general knowledge about DPP-4 and the potential pro-tumorigenic mechanisms of clinically relevant DPP-4 inhibitors. Preclinical data, including those from the authors, are well presented in the first sections of the article, followed by clinical evidence and by a very interesting section addressing the relationships between DPP-4 inhibitors and metformin, which might be a key for a better understanding of this problem. The scope of the paper is adequate, with enough depth both for the biological and the clinical aspects of the problem. The article is well organized and contains relevant and updated information potentially useful both for physicians treating diabetes and cancer. The conclusion, however, does not entirely reflect the content of the previous exposure, as further commented below.

Major comments:

- Adding a table with the data about DPP-4 expression in cancer might be of help to clarify the section 3.

- The exposition of clinical data about the association of DPP-4 inhibitors and risk of cancer seems somewhat unbalanced. The statement of lines 230-231 about the “highly controversial” status of the knowledge concerning cancer risk is not completely supported by published data since the three metanalyses cited by the authors (with over 100.000 patients) have the same conclusion of no increased risk. This conclusion is further supported by other similar analyses, also focused on digestive or pancreatic tumors. The reference to “heterogeneity of conclusions by clinical analysis” is also unsupported (no additional references are provided in the paragraph). This does not invalidate the discussion about the methodological biases and limitations of the existing data (lines 240-245), but it should be clearly stated that the current knowledge does not support an increased risk of cancer among DPP-4 inhibitors users.

- A more in-depth discussion of the potential relevance of glucose control for cancer prognosis is lacking. Although this issue is addressed on the section 5.1 (lines 238-244) dedicated to cancer risk, no mention is done in the section 5.2, which is focused on cancer progression. Glucose control has been recognized as a confusion factor also for cancer metastasis and progression in recent articles (Vihervuori, Cancer Epidemiol Biomarkers Prev 2021; DOI: 10.1158/1055-9965.EPI-19-0580) and this possibility could be included in the text.

- Concerning the potential impact of metformin on breast cancer progression or response to treatment (lines 278-284), the exposition of clinical data should be balanced with other articles showing the lack of effect of metformin on survival or response, such as a recent randomized phase II trial in the neoadjuvant setting (Pimentel et al, Breast 2019; DOI: 10.1016/j.breast.2019.08.003Metformin phase II) or a more recent metanalysis (Wu et al, Ann Transl Med 2020; DOI: 10.21037/atm-20-4441).

- The conclusion is somewhat unbalanced, focusing on the potential risks of DPP-4 inhibitors for cancer diabetic patients, but, in this reviewer’s opinion, it should be clearly stated that we have overall high-level evidence of no increased cancer risk with DPP-4 (also for pancreatic cancer) and that the main problem is the lack of adequate data on diabetic patients with cancer concerning recurrence or progression.

- Finally, regarding English language, moderate changes are required. The articles are frequently omitted and there are occasional concordance problems, although the article is overall well written.

Specific comments:

- The mTOR pathway is a key oncogenic pathway for breast cancer (especially for endocrine-resistant breast cancer) and for other tumors. Further comments on the activation of mTOR pathway by DPP-4 inhibitors (line 159) might be relevant.

- In the paragraph related to chemotherapy resistance (lines 161-171), the inclusion of androgen deprivation therapy (an endocrine-based therapy) is not appropriate.

Reviewer 2 Report

In this review the authors Kawakita et al., described the role of CD26/DPP-4: Type 2 Diabetes Drug Target with in Cancer Biology.

I found the review well written and organised. I have just some comment and suggestions:

1) may the authors add a section describing the Dipeptidyl peptidase family and their potential role in cancer. This section will provide to the reader a better understanding about this class of genes.

2) Table are too small. Please organise the tables better and increase the font size.

3) The authors must to provide a more detailed table in which current clinical trials investigating the inhibitors of D26/DPP-4: Type 2 are listed and discussed.

4) I suggest to change the titles of section with more appropriate titles.

For e.g. the authors entitled the section 4 with following title: DPP-4 inhibition on existing tumor

It should be more appropriate to use the title Strategies targeting or inhibiting DDP-4 in cancer. I strongly encourage to amend also other titles.

Reviewer 3 Report

In this manuscript, the authors reviewed the impact of DDP-4 inhibitors, anti-diabetic drugs, in T2DM patients with cancer. The expression of DPP-4 in tumor samples, the effect of DDP-4 inhibitors on tumor progression, and the effect of DPP-4 inhibitors on cancer incident and cancer prognosis in clinical studies were reviewed and discussed. Finally, the effect of co-prescription of metformin with DPP-4 inhibitors on cancer metastasis were briefly discussed, providing other choice for the management of T2DM patients with cancer. The topic discussed here is important but complex. Overall, this manuscript is relatively comprehensive. Few questions are listed below.

  1. In addition to metformin, are there other drugs that can alleviate DPP-4 inhibitor-induced cancer metastasis?
  2. It is suggested that the author briefly discuss whether there are other drugs that can be used as substitutes for DPP-4 to avoid DPP-4 inhibitor-induced cancer metastasis.
  3. Page 53 and 54: the authors need to cite references that “mentioned favorable effects of DPP-4 inhibitors 53 for cancer biology”.
  4. Figure 1: It is suggested that the authors separate the figure into two panels for clarity: the condition of tumor without DDP-4 inhibitor, and the condition of tumor with DDP-4 inhibitor.”
  5. Table 1: It is suggested that the authors separate “Consequences of processing by DPP-4 (receptor recognition)” into two columns: “receptor recognition” and “Consequences of processing by DPP-4”.
